# Tuning of Liver Sieve: The Interplay between Actin and Myosin Regulatory Light Chain Regulates Fenestration Size and Number in Murine Liver Sinusoidal Endothelial Cells

**DOI:** 10.3390/ijms23179850

**Published:** 2022-08-30

**Authors:** Bartlomiej Zapotoczny, Karolina Szafranska, Malgorzata Lekka, Balpreet Singh Ahluwalia, Peter McCourt

**Affiliations:** 1Vascular Biology Research Group, Department of Medical Biology, University of Tromsø (UiT), The Arctic University of Norway, 9010 Tromsø, Norway; 2Department of Biophysical Microstructures, Institute of Nuclear Physics, Polish Academy of Sciences, 31-342 Kraków, Poland; 3Department of Physics and Technology, University of Tromsø (UiT), The Arctic University of Norway, 9010 Tromsø, Norway

**Keywords:** fenestration, liver sinusoidal endothelial cells, myosin regulatory light chain, structured illumination microscopy (SIM), scanning electron microscopy (SEM), ROCK, MLCK, actin, MLC phosphorylation, non-muscle myosin II

## Abstract

Liver sinusoidal endothelial cells (LSECs) facilitate the efficient transport of macromolecules and solutes between the blood and hepatocytes. The efficiency of this transport is realized via transcellular nanopores, called fenestrations. The mean fenestration size is 140 ± 20 nm, with the range from 50 nm to 350 nm being mostly below the limits of diffraction of visible light. The cellular mechanisms controlling fenestrations are still poorly understood. In this study, we tested a hypothesis that both Rho kinase (ROCK) and myosin light chain (MLC) kinase (MLCK)-dependent phosphorylation of MLC regulates fenestrations. We verified the hypothesis using a combination of several molecular inhibitors and by applying two high-resolution microscopy modalities: structured illumination microscopy (SIM) and scanning electron microscopy (SEM). We demonstrated precise, dose-dependent, and reversible regulation of the mean fenestration diameter within a wide range from 120 nm to 220 nm and the fine-tuning of the porosity in a range from ~0% up to 12% using the ROCK pathway. Moreover, our findings indicate that MLCK is involved in the formation of new fenestrations—after inhibiting MLCK, closed fenestrations cannot be reopened with other agents. We, therefore, conclude that the Rho-ROCK pathway is responsible for the control of the fenestration diameter, while the inhibition of MLCK prevents the formation of new fenestrations.

## 1. Introduction

The vascular endothelium constitutes a functional barrier between the circulating blood and peripheral tissues. Endothelial cells are highly diverse, serving different functions depending on their location in the organism. Located in the liver capillaries, liver sinusoidal endothelial cells (LSEC) provide highly effective filtration functions, allowing for controlled bidirectional transport of macromolecules and solutes. The absence of a basement membrane and the presence of sub-micrometre-sized fenestrations lacking diaphragms give rise to the extreme hyperpermeability of LSEC [1], making LSEC the most permeable endothelial cells in mammals [2,3]. The passive transport of solutes through LSEC is primarily facilitated by fenestrations, which are 50–350 nm diameter, transcellular pores gathered in sieve plates. Fenestrations are present in the flat regions of the cell, occurring from the perinuclear cell zone to its periphery [4,5].

It is widely accepted that fenestrated morphology is associated with healthy LSEC. Loss of fenestrations (defenestration) is a hallmark of liver pathology [6], or aging [7,8], leading to impaired transport of lipoproteins [9] and affecting drug retention in systemic circulation [10]. Recently, it was presented that pharmacological interventions could potentially restore fenestrated morphology in partially defenestrated LSEC from old mice [8]. Therefore, the understanding of the mechanisms responsible for regulating fenestrations in LSEC is the subject of intensive research. Still, after over 50 years from the first precise description of fenestrations in 1970 [11], our knowledge about the cellular mechanisms regulating LSEC porosity (fenestration size and number) remains limited.

Data on isolated LSEC provide in-depth insights into the structure, dynamics, and regulation of fenestrations. During the last few decades, several crucial discoveries were made about cellular regulation of fenestrations [12]. Fenestrations were shown to be inducible structures—their number and diameter could be altered using chemical stimulation [13]. Several actin-targeted marine natural toxins were shown to increase porosity minutes after administration [14,15,16]. In particular, the treatment with cytochalasin B was reversible within an hour by removing the agent [13]. Many other agents were also reported to affect LSEC porosity [12]. Their effects go hand in hand with the simultaneous remodelling of the actin cytoskeleton. Besides actin, membrane scaffolds were reported to have a role in preserving the fenestrated morphology of LSECs [17,18]. Unfortunately, the complete composition of fenestrations invariably remains an enigma, which hinders the development of precise pharmacological intervention strategies to restore the porosity in LSEC. 

The existing four hypotheses describing the regulation of fenestrations in LSEC (summarised in [12]) indicate that phosphorylation of myosin might be the core of the cellular mechanisms regulating the size and number of fenestrations. In their seminal paper, Braet and Wisse reviewed in 2002 in-depth that calcium-calmodulin phosphorylation of myosin light chain (MLC) via MLC kinase (MLCK) was most likely responsible for fenestration contraction [19]. This idea was based on numerous studies about calcium-mediated serotonin effects on fenestrations by the research groups led by Arias and Oda et al. [20,21,22]. Nevertheless, the reverse effect of dephosphorylation and fenestration enlargement remained speculative [4]. Further, the Rho-ROCK pathway, which affects MLC dephosphorylation via MLC phosphatase (MLCP), was discussed as a regulator of LSEC porosity by Yokomori and later Venkatraman and Tucker-Kellogg [23,24]. The authors reported that thrombospondin-1 (TSP-1) causes defenestration by activating Rho-ROCK pathway. In addition, inhibition of ROCK via Y27632 resulted in increased porosity [24]. 

In general, the hyperpermeability of the endothelium was shown to be controlled by the non-muscle myosin II [25]. The contractile force of myosin ATPase can be activated by MLCK- or ROCK-induced Ser19 phosphorylation of MLC. Phosphorylation at Thr18 was mainly reported on stress fibres and linked to increased contractive ATPase activity. The contraction of myosin was reported to be dependent primarily on one of the pathways of MLC phosphorylation, namely via ROCK/MLCK [26]. The authors provided a detailed description of ROCK/MLCK regulation of MLC phosphorylation in endothelial cells. Both pathways were described to have an important, distinctive regulative role, and both were investigated in parallel. It was concluded that endothelial contraction leads to the opening of endothelial cell junctions and increased endothelial permeability [27]. In LSEC, however, the permeability is considered not to be regulated by opening cell junctions but by the size and number of open fenestrations. 

In the present study, we fill the gap in the knowledge about the regulation of fenestrations by phosphorylation of MLC. We show that in LSEC, both phosphorylation and dephosphorylation of MLC results in an altered fenestration number and diameter. We analyse the effects elicited by various inhibitors on LSEC, describing parameters such as porosity (the fraction of cell area covered by fenestrations), fenestration size (distribution of fenestration diameter), the distribution and shape of long fibres and actin mesh, and the level and localisation of monophosphorylated (pMLC) and diphosphorylated (ppMLC) myosin regulatory light chain. To better understand the correlation between pMLC and actin cytoskeleton, we used blebbistatin to prevent the contraction of activated myosin. Moreover, to connect the role of MLC with actin cytoskeleton, we used an actin depolymerising agent, cytochalasin B, alongside the inhibitors. To quantify the morphology of LSEC, we used modern high throughput scanning electron microscopy (SEM), a well-established microscopic technique in LSEC morphology research. We also provide data acquired with structured illumination microscopy (SIM), a method with resolution better than those imposed by the diffraction limit of light, that would allow us to colocalize pMLC and ppMLC to sieve plates and fenestrations. The selected methodology and data analysis methods were carefully selected based on our recent reports [28,29]. We supported our data with the assessment of LSEC functions, by performing the following: a viability test (lactate dehydrogenase (LDH) assay), a mitochondrial activity test (resazurin assay), and an assessment of endocytic activity (FSA scavenging assay). We conclude that the regulation of pMLC/ppMLC is divergent at ROCK and MLCK and controls not only the size but also the number of open fenestrations in LSECs.

## 2. Results

In order to unravel the role of MLC phosphorylation in the regulation of fenestrations, we firstly labelled mono- and diphosphorylated MLC (pMLC and ppMLC) in LSEC and studied the cells with SIM (Figure 1).

The freshly isolated LSEC in the primary culture display heterogenous morphology. SIM provides the super-resolution necessary for the selection of well-spread and well-fenestrated LSEC. Moreover, it enables the co-localization of the antibody with the cytoskeleton, fenestrations, and sieve plates. We showed that pMLC/ppMLC colocalize with the actin filaments surrounding sieve plates. 

We did not detect any antibody signal within the sieve plates, neither in the control nor in LSEC treated with inhibitors. In non-fenestrated areas of LSEC, p/ppMLC was uniformly distributed and colocalized with actin filaments. Moreover, ppMLC was additionally observed in the nuclear region of all treated and control samples. The use of SIM has an additional advantage—it allows not only the assessment of differences in phosphorylation levels but also provides a precise intracellular distribution of pMLC/ppMLC. In particular, after CytB treatment, we observed that the distribution of the antibody is no longer uniform, but rather presented in grouped dots, which corresponds to actin dots (Appendix A). In contrast, after CalA treatment, we observed good alignment of the antibody with actin stress fibres (Appendix A).

### 2.1. Modulation of MLC Phosphorylation

To test the hypothesis that the phosphorylation of MLC is involved in the regulation of fenestrations and to decouple the mechanism regulated via either ROCK or MLCK pathways, we divided our experiments into two groups: first, involving inhibitors affecting ROCK/MLCP and second, involving inhibitors affecting MLCK in a calcium-dependent or independent manner. Next, we presented the effects induced by the non-muscle myosin II ATPase inhibitor, blebbistatin.

#### 2.1.1. ROCK Pathway

It has already been presented that the inhibition of ROCK results in an increase in porosity [24]. Here, we repeated the experiment, searching for other aspects of LSEC morphology and functions. In addition to ROCK inhibition by Y27632, we tested if CalA has an opposite effect on the phosphorylation of MLC. Treatment with both inhibitors had profound effects on LSEC morphology. Y27632 (10 µM) caused a 20–27% increase in the porosity of LSECs after 60 min (*n* = 3) and shifted the distribution of fenestration diameters towards smaller values (Figure 2).

The proportion of fenestrations > 200 nm in the overall fenestration distribution decreases from 17% to 12% and 8% for the control, Y27632, and Y27632+CytB-treated cells, respectively. The mean fenestration size calculated from the maximum of the peak was 145 ± 52 nm for the control and 145 ± 47 nm for Y27632-treated cells. To enhance the effect of the inhibitor, after 30 min of treatment with 10 µM Y27632, 21 µM CytB was added for another 30 min. The combined effects of the two agents resulted in the additional increase of the LSEC porosity of 1.35–1.75-fold (3.4–5.1 percentage points) of the control (*n* = 3). Moreover, the diameter of the fenestrations was significantly smaller. The peak was shifted, by >25 nm, to 118 ± 45 nm. We observed broadening of the distribution of the fenestrations’ sizes towards smaller values. The detection of fenestrations in the range of 50–90 nm was reported to be overburdened with the error arising from the quality of SEM images [28]. Here, however, we were able to clearly distinguish the second group of fenestrations < 100 nm. We observed actin depolymerisation after treatment with Y27632 (Figure 3).

Moreover, the reduced tension of actomyosin caused the effect of rounding of those actin fibres that remained polymerized, which was not observed in the control. The same effect was observed after treatment with blebbistatin (see Section 2.2). The curved actin fibres became boundaries of new sieve plates after a challenge with both blebbistatin and Y27632. The combined effect of Y27632 and cytochalasin B resulted in severe depolymerisation of the actin. Porosity is a quadratic function of the fenestration diameter, so, for example, a double increase in fenestration diameter results in a four-times increase in porosity. Here, we observed a major reduction in fenestration diameter with a subsequent increase in the porosity, which indicates a significant increase in the fenestration number. The small fenestrations observed after treatment and the cumulative effect on the porosity of Y27632+CytB treatment resulted in flat and hyper-fenestrated LSEC (Appendix A). Both pMLC and ppMLC were significantly reduced after 60 min of 10 µM Y27632 treatment. The residual signal of the antibody was located on remaining actin fibres or actin dots in the case of both CytB treatment alone and in combination with Y27632. 

The opposite effect was observed after treatment with CalA (Appendix A). With a small dose of 1 nM, this inhibitor caused a 20–60% decrease in the porosity and a significant increase in the size of fenestrations after 30 min. Moreover, the second peak of 205 ± 62 nm appeared. The contribution of fenestrations larger than 200 nm changed from 19% for the control, to 51%, and 67% for CalA 1 nM, and CalA 100 nM + CytB, respectively. In higher concentrations of 10 nM and 100 nM, without CytB pre-treatment, fenestrations almost completely vanished due to cell rounding (Appendix A), hampering the evaluation of fenestration size distribution. Values of pMLC and ppMLC were significantly elevated. The tension caused by increased MLC phosphorylation caused LSEC rounding. Actin polymerisation was increased with prominent stress fibres being present throughout the cell body after a challenge with 100 nM calyculin A. To observe the effect of higher concentrations of CalA on fenestrations, we pre-treated LSEC with CytB for 30 min to reduce the formation of actin fibres (thickened actin fibres increase the cell height, disabling the formation of fenestrations [18]). The effect of CytB partially protected LSEC fenestrations from the effect induced by 10 and 100 nM of calyculin A. The tension of actomyosin was reduced due to the shortening of actin fibres. As a result, LSEC remained flat, and fenestrations were still observed. The porosity decreased and fenestration diameter enlarged from 161 ± 47 nm to 221 ± 66 nm. Because of the round shape of the holes and their grouping into sieve plates, we considered them as fenestrations, not gaps. We set the upper limit of fenestration size to 350 nm, but we could observe structures resembling fenestrations of up to 600 nm. The protective effect on fenestrations was not observed when LSEC were pre-treated with CytB prior to challenges with other inhibitors—ML-7, KN93, nor Y27632. Interestingly, a similar effect of large fenestrations and a partially fenestrated morphology of flat LSEC were still observed after pre-treatment with 20 µM of blebbistatin (Appendix A).

#### 2.1.2. MLCK Pathway

The effect of MLCK on LSEC has been previously reported to be dependent on calcium. However, some reports indicate that calcium can also affect the Rho-ROCK pathway [30]. To evaluate the effect of MLCK pathway alone, we selected two MLCK inhibitors, namely ML-7 hydrochloride and KN93 (Figure 4). The first compound is a widely used direct inhibitor of MLCK. The second is a calcium-calmodulin inhibitor that has an inhibiting effect on MLCK via calcium-dependent MLC phosphorylation. As a result, KN93 blocks only the phosphorylation of MLC in a calcium-dependent manner, while ML-7 inhibits MLCK down in the signalling pathway, affecting both calcium-dependent and independent signalling branches.

ML-7 had a dose-dependent effect on both LSEC porosity and fenestration diameter. A significant effect of the ML-7 on the porosity was observed after 60 min of treatment with 5, 10, and 20 µM, while 1 µM ML-7 had no significant effect on LSEC porosity. The highest ML-7 concentration (20 µM) caused almost a complete loss of fenestrations, without any visible effect on the level of actin polymerization (Figure 3). After 20 µM, we observed that cell–cell contact was loosened and gaps between cells occurred. The porosity and distributions of fenestrations’ diameters were calculated for those cells that remained partially fenestrated; however, most of the cells became defenestrated (Appendix A). The distribution of fenestration diameters increased by 40 nm for the maximally used concentration. The proportion of fenestrations > 200 nm changed from 7% for the control up to 38% for 20 µM ML-7. The effect became significant from 5 µM (1 µM ML-7—7%, 5 µM ML-7—15%, 10 µM ML-7—24%). ML-7 is an ATP competitive, reversible inhibitor of MLCK; thus, its effect on fenestrations should be reversible. To test this, we treated LSEC with 10 µM of ML-7 for 60 min followed by removing the inhibitor by rinsing with a fresh medium and fixation after 2 further hours in culture. Indeed, we observed that the porosity, the distribution of fenestration diameter, and the proportion of fenestrations > 200 nm (now, 14%), all (mainly) returned to the control levels. In other experiments, we combined ML-7 with CytB. Firstly, we treated LSEC with ML-7 for 30 min, followed by the addition of 21 µM CytB for the next 30 min. The effect of CytB was completely abolished and the ML-7-induced defenestration was not reversed with CytB. To show whether the inhibition of MLCK prevents the formation of fenestrations, we combined its effect with CytB, KN93, Y27632, CalA, and blebbistatin. In all cases, cells remained defenestrated (data not shown).

To verify whether the effect of MLCK inhibition on LSEC defenestration is calcium-dependent or independent, we performed experiments with the calcium-calmodulin inhibitor KN93. Treatment with KN93 at 10–20 µM has been reported to elicit relaxation of non-muscle myosin II in other types of cells [31,32]. Preliminary data obtained in this study showed no effect below 20 µM. In our hands, KN93 generated a 0.6–0.7-fold decrease in the porosity of LSEC relative to the control (a decrease of 1.0–2.5 percentage points) (*n* = 3), but its effect was not as prominent as ML-7, and never reached a complete loss of fenestrations. We observed insignificant trends towards the reduction of fenestration diameter (ctrl—130 nm, KN93—121 nm). In contrast to ML-7, the combined effect of 20 µM KN93 and CytB (30 min + 30 min) was the same as with CytB alone. Moreover, we did not observe changes in the level of actin polymerisation after KN93 treatment (Figure 3).

### 2.2. Actomyosin Contraction Uncoupling p/ppMLC with Blebbistatin

It was postulated that actin-myosin contraction can regulate fenestration diameter [12,19]. Results presented above indicate a regulation of fenestration size in an MLC phosphorylation-dependent manner. To uncouple the contractile effect of myosin on actin, we used blebbistatin, a myosin-II inhibitor. Blebbistatin was reported to affect the contractility of myosin heavy chains via ATPase activity, with less of an effect on phosphorylated MLC [33]. Our results confirm this (Figure 5). The levels of pMLC and ppMLC were similar to the control. Blebbistatin should decrease the contractile force similarly to Y27632 and ML-7, but its effect would be independent of MLC phosphorylation. We observed dose-dependent effects on actin shape after 60 min of treatment with 5 and 20 µM blebbistatin, resembling the effect of Y27632 (Figure 3). Actin fibres became rounded, forming structures similar to sieve plates with new fenestrations identified within them, and the mean fenestrae diameter was reduced by 15 nm. Although this trend follows the effect of Y27632, it is on the limit of significance (see, Methods). Still, CytB had an effect on LSEC pre-treated with blebbistatin (Appendix A). We observed no significant changes in porosity.

Blebbistatin is purely a myosin-II inhibitor and it was shown that other isoforms of myosin remained unaffected after treatment [34]. The insignificant changes in the number and size of LSEC fenestrations after blebbistatin treatment might indicate that myosin types other than myosin II are involved in the regulation. There are 18 classes of different myosins and most cells express numerous myosin isoforms with separate localizations and functions [35]. Two of them deserve to be highlighted, namely myosin II and myosin VII. The first is the most abundant and responsible for binding to actin. It is involved in the formation of stress fibers. The regulation of myosin II activity is known to involve the phosphorylation of MLCs, which increases the Mg^2+^-ATPase activity of myosin II motor domains [36]. On the other hand, myosin VII has been reported to be a calcium/calmodulin-dependent protein, decreasing its rigidity (changing conformation) in response to increased calcium concentrations [37]. Moreover, it has two FERM domains, allowing for directly or indirectly (via membrane scaffold) binding to cell membrane, making it a good candidate to regulate fenestration size. Bhandari et al. recently reported transcriptomic data of rat LSEC [38] showing a high expression of MYO7a, a gene of the myosin VIIa heavy chain. The occurrence of the myosin VII heavy chain corresponds well with the observed effects of pMLC and ppMLC phosphorylation (presented in Figure 1), remaining at the edges of sieve plates rather than inside them (Figure 6).

## 3. Discussion

The phosphorylation of MLC is regulated by MLCK and MLCP. MLC was reported to be involved in the permeability of endothelium, both in macro- and microvascular systems [26,39,40]. In general, the higher phosphorylation of MLC means greater contraction and increased permeability [27]. The permeability of LSEC (realized through fenestrations) is different from the uncontrolled permeability when cell–cell junctions are disrupted. It is believed that well-fenestrated (permeable) LSEC have relaxed actin with limited stress fibres [8,12,18]. Although the regulation of LSEC permeability in vivo is poorly understood, there are several mechanisms proposed based on the in vitro research. We recently highlighted the importance of the MLC phosphorylation in fenestration regulation. Phosphorylation of MLC has an effect on actin polymerization and, together with membrane proteins and lipid rafts, may tune the LSEC permeability. Still, reports of direct effect of MLC phosphorylation on LSEC morphology are lacking. Recently, a co-dependence of ROCK and MLCK in endothelial cells’ barrier dysfunction was presented [26], where distinctive roles of MLCK and ROCK in pMLC and ppMLC phosphorylation were highlighted. Here, we showed specific mechanisms of ROCK and MLCK to be involved in the regulation of the diameter and number of fenestrations in LSEC in vitro. Moreover, we detected the p/ppMLC at the edges of sieve plates rather than at the scaffold of individual fenestrations. We postulate that the regulation of fenestration size is facilitated by the contraction/relaxation of the cytoskeleton surrounding the sieve plates, first indicated in [41]. Fenestrations exist within the mesh-like structure comprised of actin that is attached to thicker actin fibres [42]. During contraction of the fibres surrounding sieve plates, the actin mesh inside stretches, thus causing the enlargement of fenestrations. In contrast, when the edges of sieve plates relax, the tension of the actin mesh loosens, causing a reduction in fenestration diameters (Appendix A).

The dilation and contraction of fenestrations were studied in the 1980s and is summarised by Braet and Wisse in 2002 [19]. Despite the limitations in microscopy at the time, the extensive research resulted in a conclusion that fenestration contraction occurs through MLC phosphorylation by MLCK. The results were based on the activation of MLCK in calcium-dependent pathways induced by serotonin, endothelin, and calcium ionophores [20,21,22]. The opposite mode of fenestration relaxation remained as speculation and was proposed to involve MLCK dephosphorylation. In 2004, it was presented that both MLCK and ROCK can affect the size of fenestrations via actin activation [23]. These authors used lysophosphatidic acid (LPA) as a ROCK stimulator and bacterial toxin C3-transferase as a Rho inhibitor. The results indicate similar effects to that of MLCK, i.e., the ROCK activation resulted in the shrinkage of fenestration diameters and vice versa. However, the authors reported that MLCK was further activated after LPA treatment, making it difficult to distinguish between MLCK- and ROCK-induced phosphorylation. In another report, ML-7 was shown to prevent the contractile effects of LPA in myofibroblasts, confirming the activation of MLCK after LPA treatment [43]. We believe that this contrast to our conclusions originates in the complex activation of both MLCK and ROCK pathways (Figure 7).

Here, we used direct MLCK inhibitors (KN93 and ML-7) to investigate sole inhibition of MLCK in a calcium-dependent and independent manner, without influencing the ROCK pathway. We showed that the inhibition of MLCK induces significant dephosphorylation of ppMLC and a reduction in pMLC after one hour. We observed a dose-dependent and reversible increase in the fenestration diameter, including an increase in the proportion of large (>200 nm) fenestrations. However, by increasing the dose of ML-7, the porosity decreased down to zero. It indicates that phosphorylated (activated) MLCK is involved in the process of fenestration formation. Previous reports showed that fenestration lifespan is ~20 min, but some fenestrations last for more than 2 h [13]. Moreover, newly formed fenestrations around “fenestrae-forming centres” are much smaller than the mean size of fenestrations [13,14]. Our results showed only large fenestrations after 1 h treatment, indicating that with the inhibition of new fenestration formations, mainly long-lasting fenestrations were observed. An amount of 20 µM ML-7 resulted in defenestrated LSEC without significant changes neither in cell morphology nor in the level of actin polymerisation (Figure 3). Moreover, the effect on fenestrations was not reversed by CytB treatment. Over the years, CytB became a gold standard in LSEC investigations. This microfilament drug induces up to 300% of the fenestration number in 15–30 min, without affecting their size. CytB was shown to induce fenestrations in LSEC that lost fenestrated morphology after 3 days in culture [44]. Here, after MLCK inhibition with ML-7, no new fenestrations could open after CytB treatment, even though actin was depolymerized (Figure 3). ML-7 is the first reported agent for which disappeared fenestrations cannot be reversed with CytB. The highest dose of ML-7 affected endocytosis efficiency and mitochondrial activity and therefore, its effect on fenestrations may be indirect. On the other hand, the ML-7 effect is reversible, as removing the agent restored the fenestrated morphology. The effect of KN93 resulted in no significant changes in MLC phosphorylation nor fenestration diameter. Similarly to ML-7, KN93 lowered the porosity, but its effect could be reversed with CytB. Calcium-dependent activation of MLCK can be realised either through calmodulin or through PKC, and KN93 only inhibits the first pathway (Figure 4E). MLCK still remains regulated via PKA, PKC, and PKG. It was reported recently that the regulation of porosity is cGMP- and nitric-oxide-dependent, which was proposed to act through PKG [8]. Those mechanisms would explain the different effects elicited by KN93 and ML-7, as the former only partially blocks MLCK and the latter completely deactivates it. Additionally, it was shown that calcium can also affect MLC phosphorylation via RhoA/ROCK pathway, explaining the observed shrinkage of fenestrations by altered MLCP activity by PKG [30]. The other possibility is the lack of specificity of inhibitors used in the setup. Inhibitors are powerful tools to investigate unknown cellular pathways. However, being small molecular probes, they can affect more than one signal transduction pathways and their effects must be interpreted carefully. To fully understand the effects of calcium on MLCK and ROCK in the regulation of fenestrations, an independent study is necessary. 

In particular, we showed that after 10 µM of Y27632 treatment, a decrease in pMLC and ppMLC occurred, with a simultaneous increase in the porosity. Actin filaments became less prominent than in the control cultures and more actin mesh was present. Existing actin filaments often became rounded and within closed actin circles, sieve plates were observed. Our results are in agreement with the results presented by Venkatraman and Tucker-Kellogg [24]. In that report, it was demonstrated that the porosity of LSEC increases by ~30% after ROCK inhibition with 10 µM of Y-27632, matching our results; however, the size of fenestrations after Y27632 treatment was not investigated in that report. The size of fenestrations is directly connected with their filtration selectivity [1]. For example, HDL/LDL have diameters below 100 nm, and chylomicron remnants are larger than 200 nm [1,45]. Therefore, what should be tested is not only the mean fenestration diameter, but also the whole size distribution that can be divided into fractions of large/small fenestrations. We observed a decrease in the fenestration diameters, especially of fenestrations larger than 200 nm, which were even more pronounced after Y27632+CytB treatment. The opposite effect on the porosity and fenestration diameter was observed for MLCP inhibition by CalA. It caused a dose-dependent decrease in the porosity. The observed effect is in agreement with that reported for trombospondin-1 (TSP-1) [24]. TSP-1 activates CD47 and was reported to decrease the porosity through activation of the Rho-ROCK pathway. However, we again observed contrasting results for the fenestration diameter after CalA versus TSP-1. We believe that the discrepancies occur because TSP-1 has its effect earlier in the signalling pathway. Between the CD-47 receptor and ROCK, several signalling branches that act on actin-binding proteins may be affected, making the overall TSP-1 effect more complex. The authors presented a partial inhibition of defenestration by TSP-1 after pre-treatment with a TGF-β1 inhibitor. Inhibitors that we used here act on proteins that are directly responsible for MLC (de)phosphorylation. We selected compounds that have a rapid and dose-dependent effect on fenestrations’ porosity and diameter. As a result, by using certain concentrations of the inhibitors (Y27632 and CalA), we were able to precisely tune the mean fenestration diameter in a wide range of more than 100 nm.

The effects of CalA and Y27632 on pMLC/ppMLC confirmed the desired influence of MLC phosphorylation; however, we could not clearly establish the effect of pMLC or ppMLC on fenestrations. Both ML-7 and Y27632 reduced the ppMLC, but only Y27632 had a significant effect on pMLC, indicating that pMLC plays an important role in the regulation of fenestration diameter. However, the regulation is probably more complex. CytB heavily increased pMLC and at the same time, had no effect on fenestration diameter, other than increasing the number of fenestrations. The results are in agreement with previous reports [15,46]. In addition to this, we showed that CytB can be used to exacerbate the effect of other drugs on fenestrations. A cumulative effect of CytB with other inhibitors augmented their effect on fenestration diameters—which decreased when combined with Y27632, increased when combined with CalA, and had no effect when combined with KN93. Therefore, we show that CytB, by affecting actin, would allow the opening of new fenestrations, but their size is still controlled by MLC phosphorylation. As mentioned above, the RhoA-ROCK pathway beside the regulation of MLC phosphorylation is also involved in the regulation of cofilin, profilin, and gelsolin [12,47]. These proteins are involved in fine-tuning of the globular actin/ filamentous actin balance. Together with membrane proteins and lipid rafts, all may be additionally responsible for a complex regulation of fenestrations. To the best of our knowledge, there are no reports on these proteins in LSEC.

We want to emphasize here that the fenestrated morphology is not a direct indicator of the condition of LSEC. By using inhibitors presented above, we could uncouple the porosity from the cell condition. We observed a decrease in viability and mitochondrial and endocytic activity after CytB (Appendix A) with a subsequent increase in the porosity (Figure 2 and Figure 4). At the same time, Y27632 had “beneficial” effects on pMLC/ppMLC relaxation and mitochondrial activity or endocytosis and demonstrated effects similar to CytB on fenestrations. 

The observed weak effect of blebbistatin on fenestrations in LSEC was unexpected. The uncoupling of the regulation via phosphorylation of MLC should induce effects similar to ML-7 or Y27632. Indeed, it was similar to Y27632, as the rounding of actin fibres is observed; however, only an insignificant trend in the reduction of fenestration diameter was observed after the treatment. Blebbistatin inhibits only one isoform of myosin-II. The observed lack of effect on fenestration number and size may indicate that other myosin isoforms are involved in the regulation of fenestrations. A good candidate is myosin VIIa, which was reported to be highly expressed in rat LSEC [38]. Having two band 4.1-ezrin-radixin-moesin (FERM) domains and four to five IQ calmodulin-binding motifs, it is an excellent candidate for a direct membrane binding, close to fenestrations and regulated by calcium [48]. Still, other myosin genes were as well highly expressed and their role in regulation of fenestrations should be further evaluated.

## 4. Materials and Methods

### 4.1. Chemicals

Several inhibitors, dyes, and chemicals were used. They are collated in Appendix A for clarity.

### 4.2. Cell Isolation

The experiments followed protocols approved by the local Animal Care and User Committees. All experiments were performed in 3–5 bioreplications with the total use of research animals of about 20 mice. LSECs were isolated using the modified protocol described in [13]. Briefly, C57BL/6 male mice were anesthetized using a mixture of ketamine/xylazine followed by liver perfusion and digestion using Liberase^TM^ (Roche, Darmstadt, Germany). Parenchymal cells were removed by differential centrifugation, and LSEC were separated using immunomagnetic beads conjugated with antibodies against CD146 (MACS, MiltenyiBiotec, Lund, Sweden). After isolation, cells were seeded according to the experiment type. LSECs were seeded on Ø13 mm glass coverslips for SIM, in standard 48-well plates for endocytosis assays, or in 16-well plates with glass bottoms (CS16-CultureWell™ Removable Chambered Coverglass, Invitrogen, Eugine, OR, USA) for SEM. All surfaces were coated with 0.2 µg/mL of human fibronectin for 10 min prior to seeding. LSECs were incubated for 3 h at 37 °C with 5% CO_2_ before adding inhibitors.

### 4.3. Inhibitors

To investigate the role of the actomyosin (actin-myosin complex) cytoskeleton in changes in the fenestration number and size, we proposed the sequential use of several widely used inhibitors [49]. The schematic representation of the inhibitors’ mode of action is depicted in Figure 8 and briefly described below:

*Cytochalasin B (CytB)* is used to depolymerise filamentous actin, and increases the number of fenestrations [15]. CytB is therefore used as a gold standard to easily induce fenestrations in LSEC (up to 300% increase in their number) within 15–60 min [13,50], without affecting the fenestration size [15,46]. We applied it to decrease the length of actin fibres. CytB thus decreases the overall force generated by activated myosin. We (pre)treated LSECs with CytB to enhance or abolish the effect of other inhibitors. We used 21 µM and 30 min of incubation in all experiments.*(−)-Blebbistatin* blocks myosin-II ATPase activity. It was reported that the addition of 20 µM blebbistatin decreased fibre tension by ~80% and the effect was independent of MLC phosphorylation [51]. Blebbistatin arrests the myosin cycle in states either detached from actin, or only weakly bound to actin [52]. We applied blebbistatin to decouple MLC-dependent contraction of actomyosin. We tested concentrations of 5, 10, and 20 µM (1 h incubation), and selected 20 µM because the effect on the shape of actin filaments was the most prominent, without significant effects on fenestrations.*ML-7 hydrochloride* is a reversible and ATP-competitive MLCK inhibitor. By blocking phosphorylation of MLCK, it blocks both calcium-calmodulin-dependent and independent MLCK-induced phosphorylation of MLC. The phosphorylation of MLC can still be facilitated by ROCK. We tested 1, 5, 10, and 20 µM (1 h incubation), observing a dose-dependent effect on fenestrations. However, the highest concentration reduced cell viability after 2 h of treatment (Appendix A).*KN93* is a cell-permeable, reversible, and competitive inhibitor of calmodulin-dependent kinase type II. It selectively blocks calcium-dependent phosphorylation of MLCK without disturbing the calcium-independent phosphorylation. KN93 was used at a concentration of 20 µM (1 h incubation).*Y27632 dihydrochloride* is a ROCK inhibitor that causes relaxation of myosin tension in two ways: by preventing the direct phosphorylation of MLC via ROCK and by preventing ROCK-dependent phosphorylation/inactivation of MLCP. MLCK still phosphorylates MLC, but MLCP is constantly active, causing an elevated rate of the dephosphorylation of MLC. Y27632 dihydrochloride was used at a concentration of 10 µM (1 h incubation).*Calyculin A (CalA)* is a potent inhibitor of protein phosphatase 1 and 2A. It inhibits MLCP and other phosphatases. Phosphorylation of MLC via ROCK and MLCK occurs normally, but MLCP does not reverse it. We tested 1, 10, and 100 nM for 30 min, as longer treatments were detrimental to cell viability.

### 4.4. Microscopy

#### 4.4.1. Scanning Electron Microscopy (SEM)

SEM was used to analyse the detailed changes in LSEC morphology as it provides high spatial resolution. SEM images were used for quantitative image analysis. The method of sample preparation was presented in detail previously [28]. Here, 10–15 SEM images of entire LSEC were analysed from 3–5 individual animals. All fenestrations on each cell were analysed (instead of just a selection of individual sieve plates) to maximise relevant data. The total number of analysed fenestrations per group varied depending on how the treatment affected LSEC fenestrations and varied from ~6000 fenestrations after treatment with CalA and ML-7 and up to ~50,000 fenestrations after treatment with Y27632+CytB. 

#### 4.4.2. Structured Illumination Microscopy (SIM)

SIM is a super-resolution technique that allows optical visualization of LSEC, with their well-fenestrated morphology. The images were acquired using a DeltaVision OMX V4 Blaze imaging system (GE Healthcare, Chicago, IL, USA) equipped with a 60X 1.42NA oil-immersion objective (Olympus, Tokyo, Japan) and three sCMOS cameras. We used 488, 568, and 642 nm lasers for excitation. Image deconvolution and 3DSIM reconstructions were completed using the manufacturer-supplied softWoRx program (GE Healthcare). Images projections were analysed. It is necessary to check cell morphology together with specific staining to exclude dead cells. After isolation, up to 5% of primary cells die after seeding, despite remaining well-attached to the surface. LDH experiments (Appendix A) indicate that the fraction of dead cells can be even higher. Dead cells, having highly phosphorylated MLC, hinder proper assessment of pMLC/ppMLC levels in a sample. To resolve this problem, we calculated the intensity of the antibody per cell area. To minimize the background error, we subtracted the signal from representative LSEC labelled with secondary antibodies only (negative control). We selected 15–20 LSEC per animal (3 mice in total) using actin and cell membrane staining in addition to pMLC staining for the selection of representative LSEC. Briefly, LSEC were fixed for 15 min with 4% paraformaldehyde for 15 min. Cells were then rinsed using phosphate-buffered saline (PBS) and permeabilized using 0.1% Triton X-100 for 2 min (longer incubation times were detrimental for CellMask staining), followed by rinsing and a 60 min incubation in 2% bovine serum albumin (BSA). Primary anti pMLC (Ser19) (Cell Signaling Technology, Danvers, MA, USA, #3671) (1:50 for 72 h at 4 °C) or anti ppMLC (Thr18/Ser19) (Cell Signaling Technology, #3674) (1:100 for 24 h at 4 °C) antibody was then added in blocking buffer (1% BSA, 0.05% Tween20 in PBS). According to the manufacturer, pMLC (Ser19) targets only those MLC that are phosphorylated at Ser19, while ppMLC (Thr18/Ser19) targets MLC that are phosphorylated at both amino acids and would not label any monophosphorylated MLC (neither at Thr18, nor Ser19). We performed the labelling according to the manufacturer’s protocol. To ensure the best labeling conditions and minimize cross reactivity, we never used both pMLC and ppMLC antibodies in the same sample. Next, a secondary antibody conjugated to AlexaFluor488 was added for 90 min at room temperature. Finally, phalloidin-Atto647N (1:200) in PBS was added for 30 min followed by 30 min of staining with CellMask Orange (1:500) in PBS. Samples were rinsed seven times for 5 min each with PBS and mounted on glass slides using Prolong Gold. Full names and catalogue numbers of all reagents are listed in Appendix A. All inhibitors were tested simultaneously to minimize variation between isolations. All SIM settings during image acquisition were identical. In several preparations of LSEC treated with 10 nM or 100 nM CalA, the antibody signal was saturated and shorter acquisition times were necessary. Values of the intensity of pMLC and ppMLC were calculated for individual cells. We observed some variations in the fluorescence intensity between untreated/control samples, but the trends were preserved between the animals/isolations. SIM allowed for semi-quantification of the phosphorylation level of MLC, discarding dead cells and for detailed localisation of the antibody within the cell.

### 4.5. Data Analysis

SEM data were analysed according to the semi-automatic, threshold-based method described previously [28]. Additionally, to avoid the appearance of the artificial “second peak” of fenestrations smaller than 80 nm in fenestration size distribution (see [28]), each image was manually checked and speckles originating in uneven thresholding were manually rejected. Usually, the distributions are presented as a single Gaussian fit. Here, especially after some of the treatments, we observed multiple peaks. We decided to apply Gaussian–Kernel density to present complex distributions after treatment. The uncertainty of the fenestration diameter measurements was estimated to be 14 nm, which is connected with the pixel size and the use of the threshold-based method [28]. In addition, there is a variance caused by sample preparation and the variance between animals. Therefore, we described the shift smaller than 15 nm as insignificant and measured no less than 5000 fenestrations per treatment group.

In addition to the standard way of presenting the fenestration distribution, we also discussed the ratio of fenestrations larger than 200 nm to all fenestrations. Because the area of fenestration is the quadratic function of their radius, the impact of large fenestrations on the porosity and filtration function of LSEC is huge. The porosity of the control groups varied between animals from 4–7%. In order to present the effect of the inhibitor relative to the control for all animals, we presented the data as the net increase/decrease in percentage points relative to the corresponding control.

Data were analysed and visualized using Origin Pro software (OriginPro 2022, OriginLab Corp., Northampton, MA, USA). SEM and SIM images were analysed using Fiji [53]. The statistical significance was calculated using a two-tailed Student *t*-test.

## 5. Conclusions

The aim of this study was to verify the hypothesis that MLC phosphorylation regulates fenestrations in LSEC. The size and number of fenestrations are strictly connected with the liver function. Small fenestrations in rabbits and too few fenestrations in chickens were associated with long chylomicron remnant retention in the blood stream and high susceptibility to atherosclerosis [54,55]. Moreover, defenestration is linked with hyperlipoproteinaemia [56]. Our intention was to provide a tool allowing someone to predict the effect of a pharmacological agent on LSEC fenestration by knowing its effect on MLC phosphorylation (effect on ROCK/MLCK activation). We concluded that the tuning of the diameter of fenestrations over a wide range of 100 nm, ranging from 120 nm to 220 nm, can be achieved with a specific dose of ROCK/MLCP inhibitors. Furthermore, in parallel with the change in fenestration size, we can regulate the LSEC porosity from 0% up to 12%. Our findings indicate that phosphorylation of MLC by ROCK and MLCK has different effects on the regulation of fenestration. We demonstrated that blocking MLCK resulted in the loss of fenestration, and this effect cannot be reversed by cytochalasin. This is the first report showing that cytochalasin could not induce new fenestrations.

## Figures and Tables

**Figure 1 ijms-23-09850-f001:**
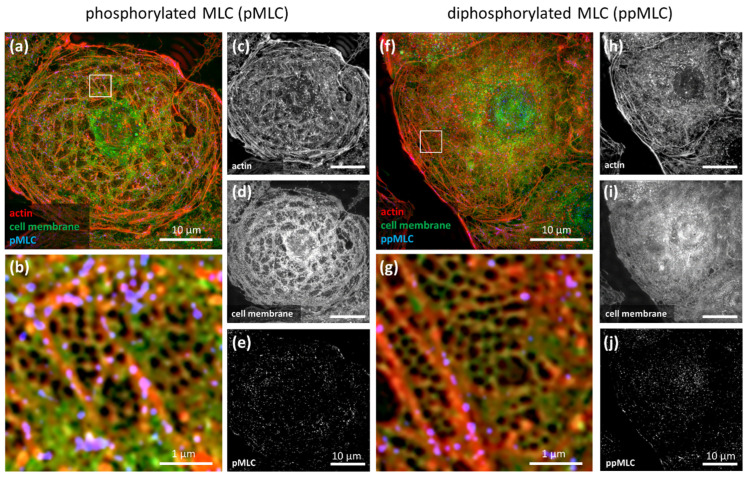
Immunofluorescence images of untreated LSEC collected using SIM. pMLC (**a**–**e**) and ppMLC (**f**–**j**). Individual black and white channels show actin (phalloidin-Atto647N, (**c**,**h**), cell membrane (CellMask Orange), (**d**,**i**), and pMCL/ppMLC staining (**e**,**j**)). Coloured images show merged channels. The selected areas of sieve plates (white boxes) were digitally magnified (**b**,**g**). The antibody is located on actin fibres at the edges of a sieve plate, and not within a sieve plate. ppMLC was also concentrated in the nuclear area. Image size: 40.96 µm × 40.96 µm, inset size: 4.1 µm × 4.1 µm. A gamma of 0.4 was used for the actin and cell membrane for better visualization of the data.

**Figure 2 ijms-23-09850-f002:**
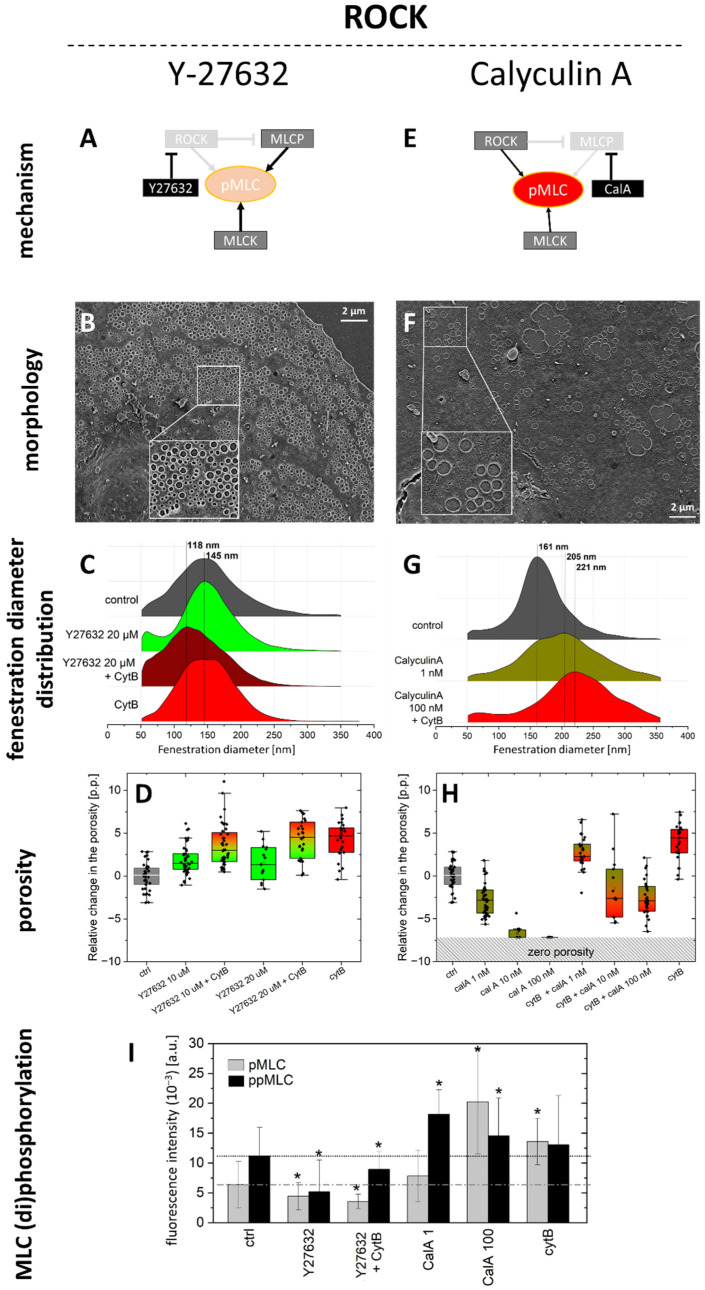
The effect of ROCK and MLCP inhibition on fenestrations in LSEC. (**A**) Schematic presentation of the mechanism of ROCK inhibition by Y27632. MLC is partially dephosphorylated (faded-red colour) by direct ROCK inhibition by Y27632. Phosphorylation by MLCK is still possible, but MLCP cannot be phosphorylated (deactivated) by ROCK, causing additional dephosphorylation. (**B**) Representative SEM image of LSEC treated with 10 µM Y27632+CytB. The inset highlights two groups of normal-sized and <100 nm-sized fenestrations. (**C**) Fenestration diameter distribution (fenestrations measured: ctrl-7843, Y27632-36358, Y27632+CytB-54492, cytB-17230). The proportion of fenestrations >200 nm is 17%, 12%, 8%, and 11% for ctrl, Y27632, Y27632+CytB, and CytB, respectively. In addition, the cumulative effect of 10 µM Y27632 with CytB resulted in the formation of a second group of fenestrations of diameters < 100 nm. (**D**) A change in the porosity relative to the control is presented (*n* = 3). The ROCK inhibition resulted in a significant increase in porosity; the effect is cumulative with CytB. (**E**) A schematic presentation of MLCP inhibition by CalA. MLC is phosphorylated (vivid-red colour) by ROCK and MLCK and no dephosphorylation pathway (via MLCP) is available. (**F**) A representative SEM image of LSEC treated with 100 nM CalA+CytB. The inset highlights fenestrations with diameters > 500 nm (presented SEM images B and F are in the same scale). (**G**) Fenestration size distribution after MLCP inhibition (the number of fenestrations measured: ctrl-6157, CalA-3467, CalA+CytB-1103). The proportion of fenestrations > 200 nm increased with the appearance of a second peak of 206 nm. Pre-treatment with CytB (30 min) resulted in mean 67% (*n* = 3) fenestration diameter >200 nm. (**H**) Change in porosity relative to the control (*n* = 3). MLCP inhibition resulted in a significant decrease in the porosity down to zero (no fenestrations identified) for 100 nM; the defenestrating effect of CalA is partially reduced by 30 min pre-treatment with CytB. (**I**) Fluorescence signal for pMLC and ppMLC measured using SIM for individual well-spread (fenestrated) LSEC. * *p* < 0.01, relative to control. The mean porosity of the control varied from 5.5% to 7.5%.

**Figure 3 ijms-23-09850-f003:**
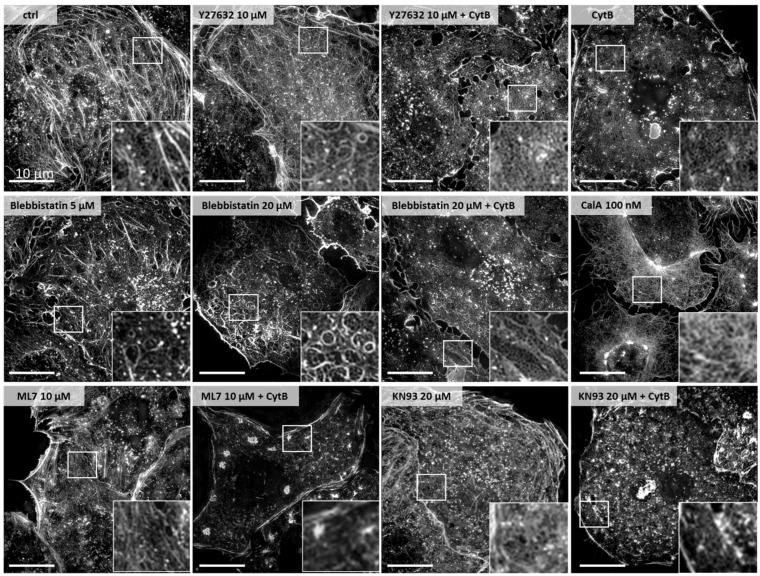
The SIM projection images (phalloidin-Atto647N actin staining) of representative LSEC treated with the inhibitors. The selected area (white square) on LSEC periphery was digitally magnified (inset). All images size: 40.96 µm × 40.96 µm, insets: 6.2 µm × 6.2 µm.

**Figure 4 ijms-23-09850-f004:**
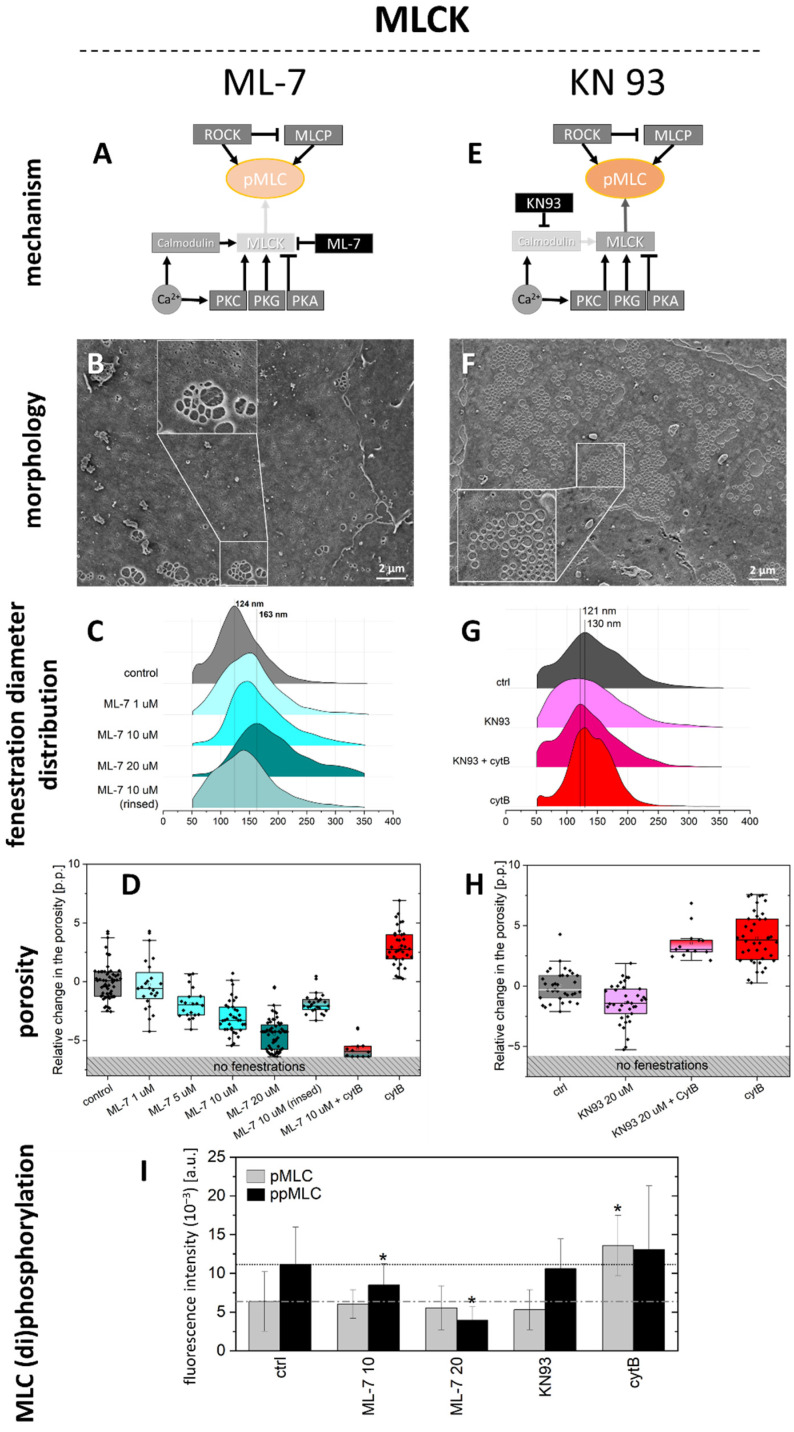
The effect of direct MLCK inhibition and calcium/calmodulin-dependent MLCK inhibition on LSEC fenestrations. (**A**) A schematic presentation of the ML-7 effect. MLC is partially dephosphorylated (faded-red colour) by direct MLCK inhibition. Phosphorylation by Rho/ROCK pathway still occurs. (**B**) A representative SEM image of LSEC treated with 20 µM of ML-7. The inset highlights a small sieve plate with enlarged fenestrations. (**C**) Fenestration diameter distribution (fenestrations measured: ctrl-22618, 1 µM ML-7-8283, 5 µM ML-7-12560, 10 µM ML-7-4630, 20 µM ML-7-870, 10 µM ML-7 (rinsed)-6607). The proportion of fenestrations > 200 nm is 7% (ctrl), 7% (1 µM ML-7), 15% (5 µM ML-7), 24% (10 µM ML-7), 38% (20 µM ML-7), and 14% (10 µM ML-7, rinsed). (**D**) Changes in the porosity relative to the control are presented (*n* = 3–6). MLCK inhibition by ML-7 resulted in a significant and dose-dependent decrease in porosity; the effect is not reduced by CytB. (**E**) A schematic presentation of calcium/calmodulin-dependent inhibition of MLCK. MLCK activation by PKC and PKG still occurs. MLC phosphorylation remains unchanged. (**F**) Representative SEM image of LSEC treated with 20 µM KN93+CytB. The inset highlights fenestrations gathered in the sieve plate, showing a similar response to the effect of CytB alone (presented SEM images are in the same scale). (**G**) Fenestration size distribution after calcium/calmodulin inhibition (number of fenestrations measured: ctrl-6341, KN93-5306, KN93+CytB-5830, cytB-35677). The proportion of fenestrations > 200 nm is similar to the control (ctrl-14%, KN93-14%, KN93+CytB-11%, CytB-4%). (**H**) A change in porosity relative to the control is presented (*n* = 3). The calcium/calmodulin inhibition resulted in a significant decrease in porosity; the effect of KN93 does not hamper the increase in porosity caused by CytB. (**I**) A fluorescence signal for pMLC and ppMLC measured using SIM for individual well-spread (fenestrated) LSEC. * *p* < 0.01, relative to the control.

**Figure 5 ijms-23-09850-f005:**
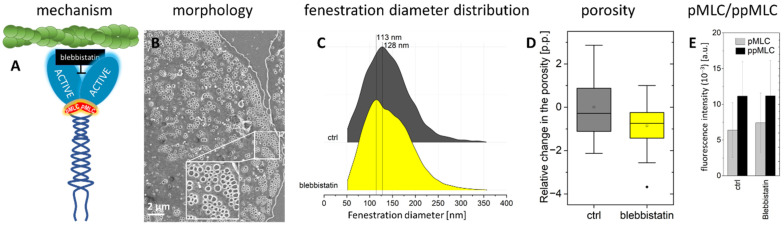
The effect of myosin heavy chain inhibition by 20 µM blebbistatin. (**A**) Schematic presentation of the blebbistatin effect—myosin cannot exert a force on actin, as the myosin motor domain is blocked. The use of blebbistatin uncouples the dependence of pMLC activation on actomyosin contraction. (**B**) Representative SEM image of LSEC treated with 20 µM blebbistatin. (**C**) Fenestration diameter distribution (number of fenestrations measured: ctrl-5402, 20 µM blebbistatin-26017). The proportion of fenestrations > 200 nm is 12% (ctrl) and 13% (20 µM blebbistatin). (**D**) A change in porosity relative to the control is presented (*n* = 2). (**E**) A fluorescence signal for pMLC and ppMLC measured using SIM for individual well-spread (fenestrated) LSEC.

**Figure 6 ijms-23-09850-f006:**
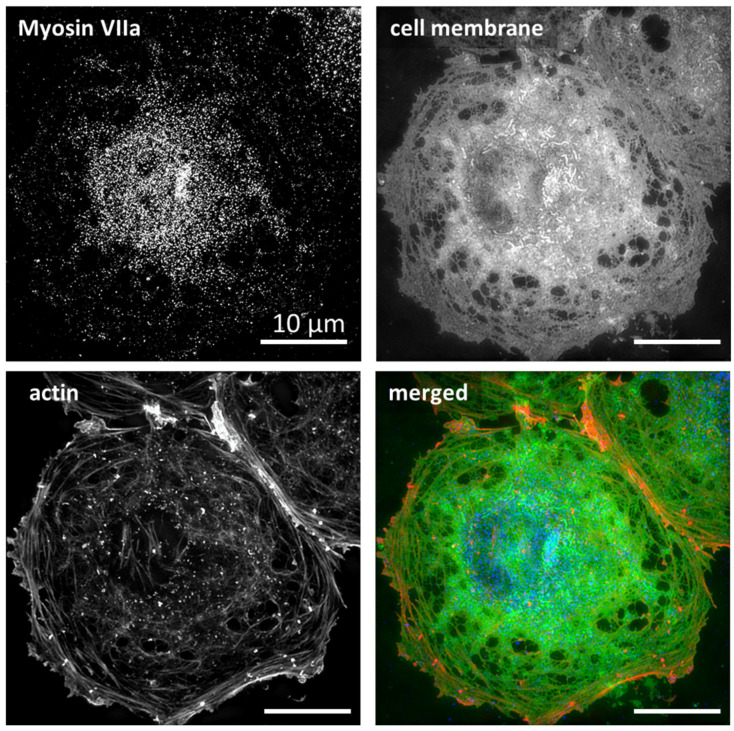
SIM projection images of a representative LSEC (control, untreated cells). Myosin VIIa/MYO7A rabbit antibody (1:100), secondary donkey anti-rabbit, Alexa 488 (1:100), cell membrane-CellMask Orange, actin-phalloidin-Atto647N. Image size: 40.96 µm × 40.96 µm.

**Figure 7 ijms-23-09850-f007:**
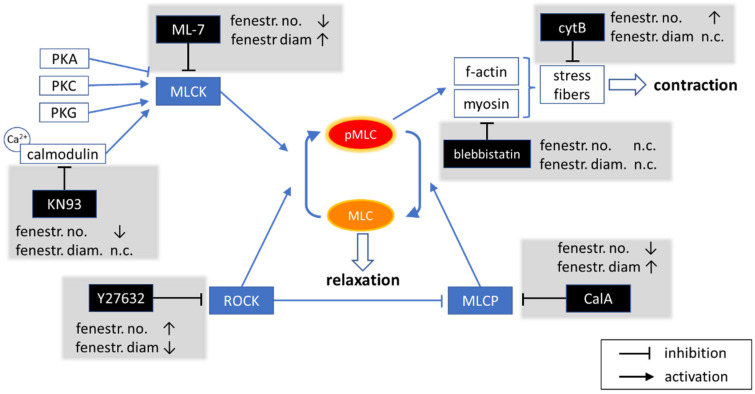
A diagram showing the effect of selected inhibitors on the regulation of MLC phosphorylation and the observed effect on fenestrations in LSEC. n.c.—no change, ↓—fenestration number (fenestr. no.)/diameter (fenestr. diam.) decrease, ↑—fenestration number/diameter increase.

**Figure 8 ijms-23-09850-f008:**
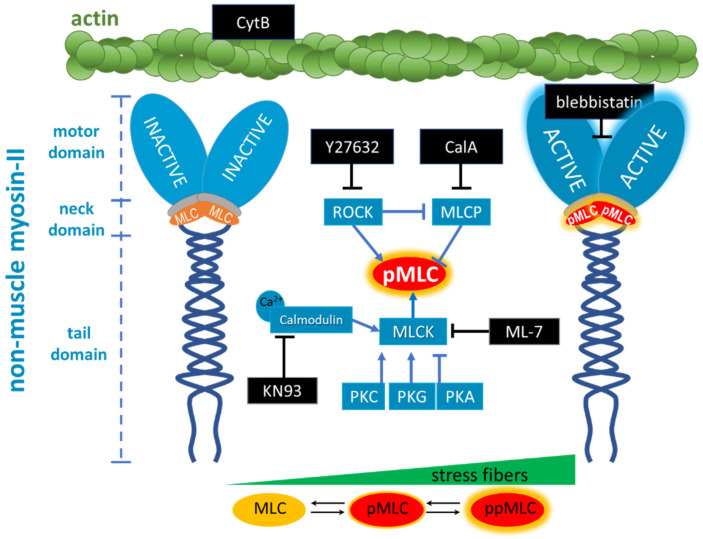
Schematic representation of the action of selected inhibitors used to test the influence of phosphorylation of regulatory myosin light chain (MLC) on fenestrations in LSEC. Phosphorylated MLC (pMLC, glowing red) activates the myosin motor domain that exerts a contractile force on actin. The contractile force generation on actin can be inhibited by blebbistatin independently of MLC phosphorylation. Cytochalasin B (CytB) disturbs actin fibres and therefore weakens the contractile force of the actomyosin complex. KN93, ML-7, and Y27632 inhibit MLC phosphorylation in different ways. Calyculin A (CalA) inhibits MLC phosphatase (MLCP) and prevents MLC dephosphorylation. At the bottom of the figure, it is also highlighted that higher levels of MLC phosphorylation (monophosphorylation (pMLC) and diphosphorylation (ppMLC)) are associated with more prominent actin stress fibres.

## Data Availability

Not applicable.

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
