# Peer review of "Tuning of Liver Sieve: The Interplay between Actin and Myosin Regulatory Light Chain Regulates Fenestration Size and Number in Murine Liver Sinusoidal Endothelial Cells"

_ijms, 2022, doi:10.3390/ijms23179850_

Round 1

Reviewer 1 Report

Zapotoczny et al. report a careful study of the impact of the kinase/phosphatase-mediated mechanisms regulating myosin light chain (MLC) phosphorylation on the dynamics of fenestrae in liver sinusoidal endothelial cells (LSEC).  Isolated LSEC were examined by scanning electron microscopy (SEM) and structured illumination microscopy (SIM) in the presence of several selective agents to determine how modulating myosin light chain phosphorylation affects the diameters and numbers of LSEC fenestrae and the fraction of LSEC surface area occupied by these pores.  Selective antibodies were used to determine mono- and di-phosphorylation of the myosin regulatory light chain, a major determinant of force development by actin-myosin interaction.  In general, increased myosin light chain mono-phosphorylation was associated with increased fenestration diameter and decreased LSEC porosity.

The text is quite coherent.  The study addresses important questions regarding the mechanisms that determine the passage of solutes both large and small between the sinusoids and the Space of Disse, from which hepatocytes extract myriad substances for detoxification, bile formation, protein synthesis and ATP production.  The topic is appropriate for this journal.  The introduction is exceptionally well-written, albeit somewhat lengthy, and very effectively brings the reader up to speed and underscores the importance of the questions addressed in the study.  Up-to-date literature is cited.  The conclusions are well-supported by the findings.  The testing of different concentrations of each agent to define the appropriate concentrations to be combined with the actin depolymerizing agent, cytochalasin B (CytB), is an important strength.

The following major concern is noted regarding the absence of some crucial information from Figure 2.  Aside from that matter, there are a few minor concerns.

In Figure 2, Y-27632 alone did not shift fenestration diameter distribution.  Only when combined with CytB did Y27632 decrease fenestration diameter.  On the other hand, CytB alone increased porosity at least as much as did CytB + Y27632 (Figure 2D), but the effects of CytB alone on fenestration diameter are not shown.   One could argue that Y27632 did not affect fenestration diameter, and the decrease in diameter in the presence of the two compounds was due entirely to CytB, i.e. was independent of Y27632-induced reduction in MLC monophosphorylation.  Was the effect of Y-27632 + CytB cotreatment simply an additive effect, or an interaction?  To resolve these questions, in Figure 2C please include the fenestration diameter distribution in presence of CytB alone.

Comparison of Figure 2G vs. 2H raises similar questions.  Because CytB alone increased porosity, one would expect CytB to lower fenestration diameter, but again the effects of CytB on fenestration diameter aren’t shown.  On the other hand, while 1 nM calyculin A increased fenestration diameter and lowered porosity, and 100 nM calyculin A alone lowered porosity to zero, 100 nM calyculin A + CytB produced even wider fenestrae than 1 nM calyculin A alone, yet the combination produced little if any change in porosity.  These results are very difficult to interpret without knowing the effects of CytB alone, or CytB + 1 nM calyculin A, on fenestration diameter.  Please provide that information.

Minor comments:

The statement on lines 23-24 is unclear.  Do you mean “after inhibiting MLCK, closed fenestrations cannot be reopened with other agents”

Line 40: Change “sized” to “diameter”

Figure 2B, F: Please indicate the treatment conditions represented in these SEM images.

Figure 3: Please add arrows and/or arrowheads to identify salient features in these images, e.g. actin polymerization/depolarization, large and small diameter fenestrae, and rounding of actin fibres.

Lines 202-203: Wouldn’t the area of a fenestration equal pi times square of the radius?

Line 224: Fenestration diameter did not double.  Area nearly doubled, increasing by 88%.

Figure 7: Several features are missing from this diagram.  Six of the boxes aren’t labeled, and there is no indication of the effects of ML-7, cytB, KN93, Y27632 or CalA on their respective targets.

Section 4.2: Please indicate the number of animals used in the study.

Lines 558-560: Wouldn’t an inhibitor of MLCP increase MLC phosphorylation?

Lines 569-570: Are these values truly the number of fenestrations per animal?  Surely a whole mouse liver has many times the numbers of fenestrations given here.

Reviewer 2 Report

This manuscript provides interesting associations between changes in ROCK and MLCK activity and subsequent alterations in fenestrations of cultured mouse liver sinusoidal endothelial cells (LSECs).  The authors infer that phosphorylation of myosin light chain (MLC) is a key regulating factor.  Overall, the paper appears to focus on an important knowledge gap, and the Introduction is well-written.  The Result section, however, suffers from a very complex delivery that makes it difficult to follow.  The figure legends are overly wordy, containing information that could be better placed in the Methods.  The overall exposition would improve with one or more sentences summarizing the overall message to be conveyed in each data section.  While the morphologic data are quite compelling, there are several issues, outlined below, that should be resolved:

[1] Antibody studies: Information about the specificity of the antibodies employed is not provided in the paper.  It is crucial to show that the antibodies directed against pMLC and ppMLC do not cross-react, as observations that rely on these reagents are the crux of the paper.  Also, the positive or negative reactivity of these antibodies with MLC under various conditions should be demonstrated using another method such as western blotting to veriufy the staining results.  In addition, it will be important to show controls for the antibody staining experiments; ideally this would mean parallel samples treated without the primary antibody or staining with an irrelevant isotype control.

[2] Inhibitor studies:  Studies presented in the Supplemental Data suggest that the inhibitors employed do not cause significant cellular toxicity in cultured LSECs at the doses utilized.  However, the authors should provide some evidence of the specificity of the reagents used. For example, is there any cross-inhibitory activity between agents used to block ROCK and those used for MLCK?

[3] The paper would be strengthened by data showing that the same pattern of change with respect to LSEC fenestration can be observed in vivo.

[4] The authors rely solely upon murine LSECs.  Complementary data on human LSECs would be a valuable addition.  To what extent do the murine cells and their fenestrations resemble those in the human liver?

[5] Original magnifications and/or scale bars should be included for all microscopy images.  In some figures, resolution appears to be suboptimal.

[6] All acronyms should be defined at first use, even in the Abstract.

[7] There is little discussion about the physiologic significance of findings.  The authors should  include some discussion about how this new knowledge is likely to enhance our understanding of liver/biliary disease. 

Round 2

Reviewer 2 Report

Regarding question 1, the authors should at least describe the evidence that supports the specificity for the antibodies used - whether it comes from the manufacturer or from their own investigations.

Regarding question 2, they should acknowledge the limitations associated with using inhibitors in the discussion of the paper.
